# Mapping the Research into Mental Health in the Farming Environment: A Bibliometric Review from Scopus and WoS Databases

Manel Díaz Llobet [1], Manel Plana-Farran [2,*], Micaela L. Riethmuller [3], Victor Rodríguez Lizano [4], Silvia Solé Cases [5,6] and Mercè Teixidó [1]

[1] GRIHO Research Group, Universitat de Lleida, Jaume II St. 69, 25001 Lleida, Spain; manel.diazllobet@udl.cat (M.D.L.); merce.teixido@udl.cat (M.T.)

[2] Facultat de Dret, Economia I Turisme (FDET), Universitat de Lleida, Jaume II St. 73, 25001 Lleida, Spain

[3] School of Population Health, Curtin University, Kent Street Bentley, P.O. Box U1987, Perth 6102, Australia; micaela.riethmuller@postgrad.curtin.edu.au

[4] Department of Agricultural Economics, University of Costa Rica, San José 11501-2060, Costa Rica; victorantonio.rodriguez@ucr.ac.cr

[5] Faculty of Nursing and Physiotherapy, Universitat de Lleida, Montserrat Roig 2, 25006 Lleida, Spain; silvia.sole@udl.cat

[6] Departamento de Fisioterapia, Universidad de Málaga, Andalucía TECH, Arquitecto Penalosa 3, 25009 Málaga, Spain

[*] Correspondence: manel.plana@udl.cat

**Abstract:** A significant part of the world economy is devoted to agriculture. The sector accounts for 27% of global employment and 4% of global GDP. Approximately 28.5 million farms are located in Europe and Latin America. In this sector, many uncertainties negatively impact farmers' mental and emotional well-being. Many factors contribute to increased stress and a worsening of farmers' mental health, including health problems resulting from the conducting their profession, economic uncertainty, the effects of climate change, and technological changes in the agricultural sector. Despite the existence of literature review studies related to mental health in agriculture, no bibliometric review study has been conducted. This article presents the first in-depth bibliometric analysis of the scientific literature on mental health in agriculture and operates based on Scopus and Web of Science databases. The results are presented as tables and explanatory diagrams describing the findings. The findings show the exponential increase in research in the last ten years and the evolution towards more social and health-related topics across the previous five years. The most common keywords are "suicide", "stress", and "depression". No topic has been found where the current scientific production was significantly larger than the rest, indicating the wide variety of research sub-topics in this field.

**Keywords:** mental health; farming; bibliometrics; agriculture; review

## 1. Introduction

The agricultural sector represents 27% of global employment [1], and farming accounts for 4% of the global gross domestic product (GDP) according to the World Bank (2020) [2]. In this vein, in Europe, figures gleaned from Eurostat show that there are around 12 million farms in the EU, and 28.96% are worked either solely by family members or see them do most of the work [3]. Similarly, there are roughly 16.5 million farms throughout Latin America and the Caribbean, and eight of ten are considered family farms [4].

This sector has experienced a revolution based on a long-lasting scheme that requires a mindset change. These changes occur as the agricultural industry rapidly changes with increasing technology, resulting in the need to invest in more expensive capital to remain efficient [5]. In addition to technological changes, the sector faces serious social, economic and environmental changes [6]. Agriculture and farmers are on the verge of important and

intense adjustments [7]. Farmers are required to produce more while remaining efficient; as a result, the size of farms increases while the number of farmers reduces, increasing the risk of social isolation as farming community populations decline [8,9].

Consequently, in line with the perspective raised above, it can be stated that agriculture represents one of the most intense scenarios in terms of sectors where transformations have taken place [10,11]. Society's demands on farming are shifting "from something that provides one good (food) to something that supplies many (food, access to green spaces, healthy rural environment, flood resilience, reduced greenhouse gas emissions)" [12].

Farmers are considered key actors in society [13]. Consequently, the present and the future of farming rely on agricultural workers, managers, and next-generation farmers [14]. In this vein, Suess-Reyes et al. [15] state that the future of farming relies on farms' adaptability to the altering environment, especially family farming. However, working on a farm is a full-time activity that is often unknown to the general population despite its importance in daily life. This agricultural work involves activities such as agriculture, horticulture or domestication and is constantly affected by unpredictable aspects such as climate change, price evolution or global market conditions [16]. Furthermore, there are other risk factors for the producers: long working hours, social isolation and long distances to medical centers [17]. As a result, there is an increasing amount of literature warning of a pandemic of mental health problems in farmers.

"The pervasiveness of mental health problems in agriculture is widely recognised as a major issue affecting the industry across several international contexts, particularly in the Global North" [18]. Although the concept of mental health problems has been widely used in different contexts, there is evidence that farmers exhibit one of the highest rates of suicide, resulting in a higher risk of developing mental health problems [16]. Therefore, this topic has received particular attention, and a comprehensive body of research has addressed the importance of studying and understanding the impact of multiple pressures and stressors affecting farmers' mental health [19–25].

Along these lines, the American National Institute for Occupational Safety and Health described how farm owners had the highest rate of deaths due to stress-related conditions over a total of 130 different professions [26]. It has been found that around 71% of farmers in the Midwest experience anxiety, and 53% experience depression [27]. Furthermore, in the United States, male farmers die by suicide at twice the rate of men in the general population [28]. In Australia, the suicides are 59% higher in farmers, with one dying every 10 days [29].

One of the first systematic reviews in this field found 167 original articles and identified key risk factors [24]. The five main risk factors were pesticide exposure, finance, weather uncertainty, poor physical health or past injuries, and farming in general (heavy workload/stress/hazards). Another review examining mental health interventions in farmers surveyed most of the literature in two countries: Australia and the United States [17]. This review defined some drivers for farmers' mental health: available and appropriate care, physical health, social support, financial well-being, coping skills and assistance during a crisis [17]. Along these lines, a qualitative study on farmers' mental health and well-being identified risk factors, including weather variability, rate of change and declining population in farming communities [9]. This study also found that a capacity for "switching off" and having good social connections were important protective factors for farmers' well-being [9].

Despite the number of reviews, there is a lack of studies about bibliometric research related to mental health in farming. There have been no bibliometric reviews based on research into mental health in the farming environment, which represents a gap that should be addressed. It is commonly accepted that bibliometric reviews provide a review of the current state of the scientific literature on a topic, its findings, and current trends. These are very important aspects for the advancement of research on the topic. As a result, this paper aims to conduct a bibliometric analysis to review the state of the literature related to mental health in farming through a bibliometric analysis of scientific articles indexed

in Web of Science (WOS) and Scopus. These two databases collect the most important scientific documents and allow for data export in a format compatible with performing the analysis presented in this work. Therefore, this paper focuses on the quantity, quality, geographic areas, authors, and subject matter of research conducted thus far on this topic. For this purpose, four research questions are posed:

RQ1: What is the quantitative and qualitative level of the scientific research conducted so far on mental health and farmers?

RQ2: Which researchers and in which geographical areas have been most investigated on mental health in agriculture?

RQ3: What are the facets of mental health in agriculture studied in the research carried out so far?

RQ4: Which are the emergent research topics related to mental health in farming?

The paper is divided into four sections. In the first section, we perform an incremental search for the keywords important to the study, following the PRISMA statements. Following this, the information obtained from the documents, tables, and figures are presented to answer the research questions. As a result of the findings gathered in this section, the discussion section can proceed to present the findings and answer the research questions posed. Finally, the conclusion section summarizes the main conclusions reached by this scientific study after the initial hypothesis has been tested for validity.

## 2. Materials and Methods

### 2.1. Data Source, Data Extraction, and Study Selection

The bibliometric analysis was conducted using the Scopus and WoS databases. The query used was based on key terms related to the topic: "mental health" and farm*. The query returned all documents with the keyword mental health and any words starting with "farm" in the title, keywords, or abstract. So, all words derived from "farm" were included in this list. The PRISMA diagram in Figure 1 shows the search process.

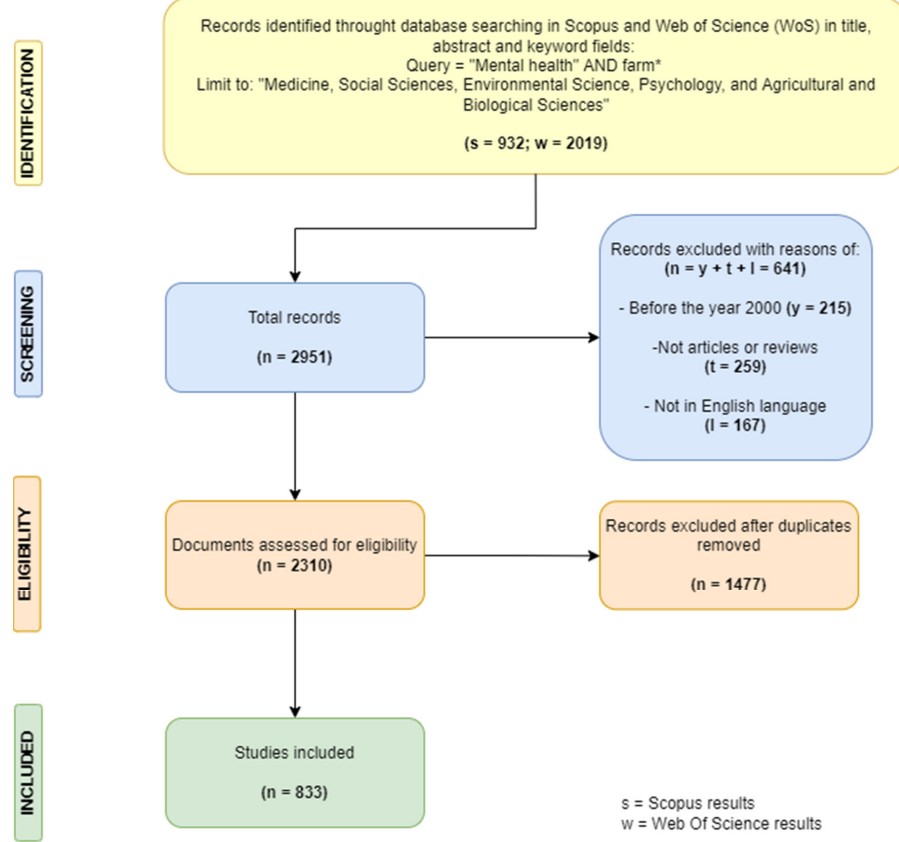

**Figure 1.** PRISMA diagram.

Our selection was based on the five knowledge areas that offer the most results: "Social Science", "Psychology", "Agricultural", "Medicine", and "Biological Sciences". These areas are shown in the "Identification" section of the PRISMA diagram (Figure 1). Several documents appeared in the first search results, confirming the topic's popularity at the scientific research level. The 2951 initial documents were filtered using the criteria described in the "Screening" section. The filter only includes articles and reviews written in English in the 21st century. Lastly, duplicates were eliminated. A total of 833 documents were used for this analysis.

### 2.2. Data Analysis and Visualization

The bibliometric analysis was carried out using VOSViewer (Version 1.6.20) [30] and RStudio software (Version 4.1.3) with a bibliometrix library [31]. Using co-occurrence data, VOSviewer can generate keyword maps, and the bibliometrix library in RStudio software offers a set of tools for quantitative data analysis. A general overview of this database is that there are 833 documents containing information from 411 sources, 2725 authors, 4014 Keywords Plus, and 1989 authors' keywords. Additionally, the database grows at a rate of 4.61 per cent per year, with an average of 18.56 citations per document. In total, 38,307 references are included in the database.

## 3. Results

### 3.1. Bibliometric Analysis

The trend in the number of publications between 2000 and 2022 is shown in Figure 2. It highlights the contrast between the small number of scientific publications at the beginning of the decade and the number of publications associated with exponential growth available at the end of the decade.

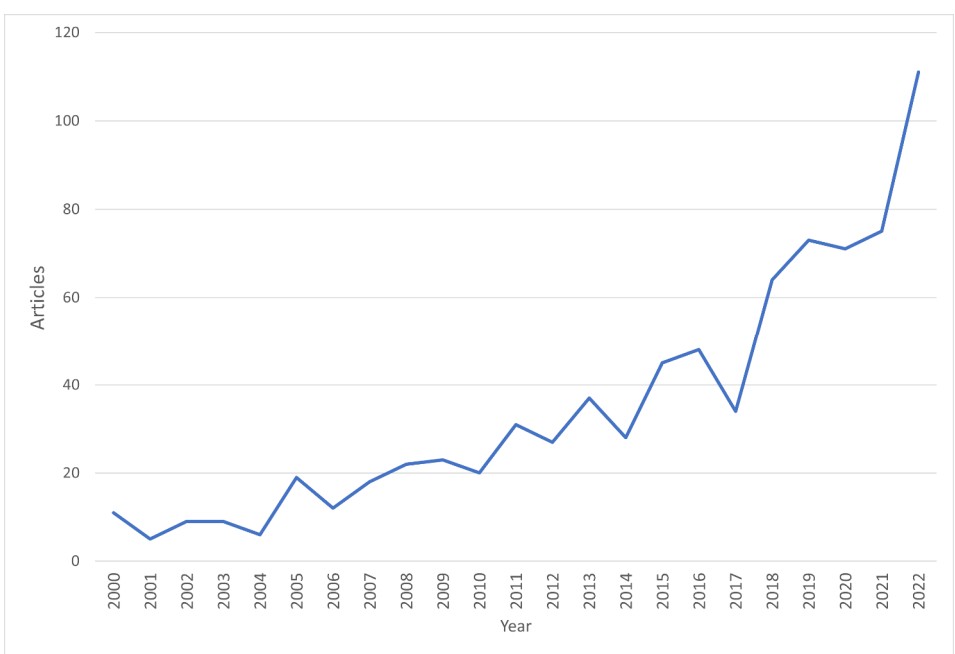

**Figure 2.** Evolution of annual scientific research on mental health and farming in the period 2000–2022 in WoS and Scopus databases using R-studio software and bibliometrix libraries.

Figure 3 illustrates a three-field plot that relates countries, affiliations, and keywords. A maximum of 20 fields were allowed in each field. The figure shows the relationship between 7 countries, 20 keywords, and 20 affiliations. The keywords "mental health", "rural", and "agriculture" were the most used keywords. Moreover, the three affiliations that published the most documents about the topic were "The University of South Australia" in Australia, "The University of Toronto" in Canada, and "Deakin University" in Australia.

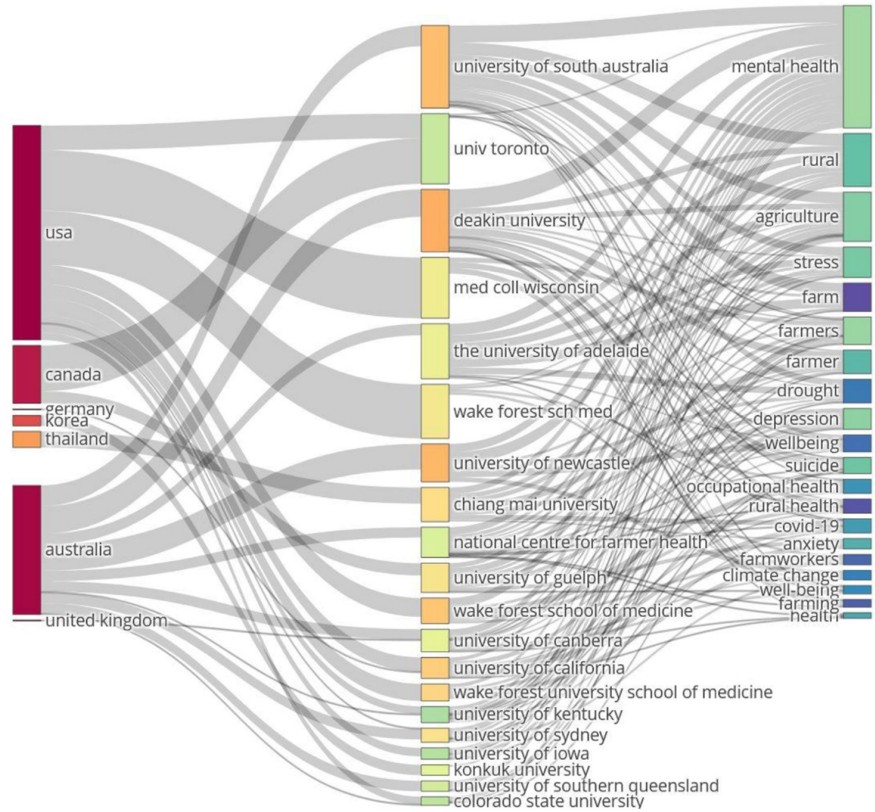

**Figure 3.** Three-field plot relating keywords, affiliations and countries using R-studio software (Version 4.1.3) and bibliometrix libraries of both databases.

Figure 4 lists the most relevant sources on the topic during the period analyzed. The source with the highest number of publications was *International Journal of Environmental Research and Public* Health, followed by *Journal of Agromedicine*, *Journal of Cleaner Production*, and *Tunneling and Underground Space Technology*.

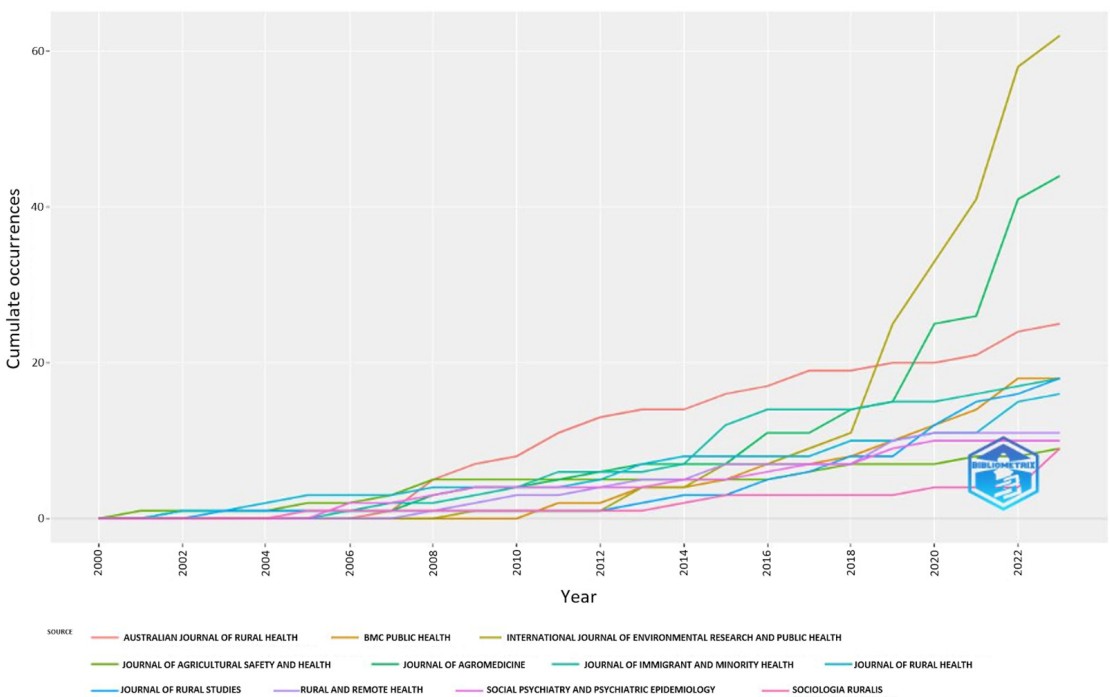

**Figure 4.** Most relevant sources using R-studio software and bibliometrix libraries of both databases.

Figure 5 presents the world's scientific production on the topic. Countries are colored in two different ways. The gray-marked countries indicate that no research related to this topic has been conducted, while the blue-marked countries suggest that scientific research and publications have been completed. It can be seen that most documents were from the USA and Australia, followed by the UK, Canada, China, and Norway.

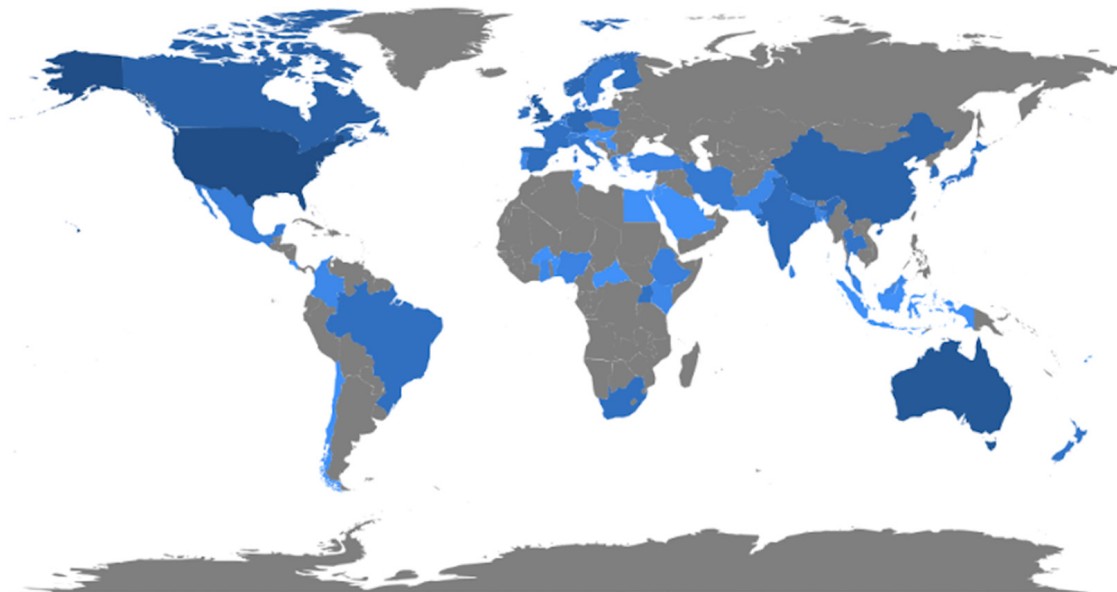

**Figure 5.** Country Scientific Production using R-studio software and bibliometrix libraries of both databases.

The top ten authors with the most publications are presented in Table 1. All the fields in the table are obtained from both databases. In terms of publications, Brumby holds the top position (h-index = 16 (S)/14 (W)), hailing from Australia at Deakin University. She is followed by Grzywacz (h-index = 56 (S)/53 (W)) and Arcury (h-index = 53 (S)/49 (W)) from the USA at Wake Forest University School of Medicine. Authors from the United States and Australia have the most publications. This table highlights that only the 10th position is reached by a German author. Based on the relationship between the authors, there are two clusters of working groups, one in each leader's country. The first cluster shows the relationship between Arcury, Grzywacz, and Quandt from the USA. The second show the relationship between Brumby, Gunn, and Kennedy from Australia, who publish together.

**Table 1.** Authors with the highest number of publications in the field of study between 2000 and 2023.

| Author | Institution | Country | Number of Publications on the Topic in Both Databases | Number of Publications Scopus (S) WoS (W) | Number of Citations Scopus (S) WoS (W) | Total h-Index Scopus (S) WoS (W) |
|---|---|---|---|---|---|---|
| BRUMBY Susan | Deakin University | Australia | 20 | 58 (S) 51 (W) | 746 (S) 632 (W) | 16 (S) 14(W) |
| GRZYWACZ Joseph | Florida State University | USA | 18 | 244 (S) 222 (W) | 12,248 (S) 9225 (W) | 53 (S) 49 (W) |
| ARCURY Thomas | Wake Forest University School of Medicine | USA | 17 | 447 (S) 464 (W) | 13,683 (S) 11,779 (W) | 56 (S) 53 (W) |
| QUANDT Sara | Wake Forest University School of Medicine | USA | 16 | 433 (S) 411 (W) | 17,481 (S) 14,531 (W) | 56 (S) 51(W) |
| KELLY Brian | University of Newcastle | Australia | 14 | 262 (S) 205 (W) | 11,881 (S) 3562 (W) | 48 (S) 29 (W) |

**Table 1.** *Cont.*

| Author | Institution | Country | Number of Publications on the Topic in Both Databases | Number of Publications Scopus (S) WoS (W) | Number of Citations Scopus (S) WoS (W) | Total h-Index Scopus (S) WoS (W) |
|---|---|---|---|---|---|---|
| KENNEDY Alison J. | Deakin University | Australia | 13 | 20 (S) 24 (W) | 232 (S) 276 (W) | 9 (S) 6 (W) |
| BERRY Helen | Faculty of Medicine and Health | Australia | 9 | 75 (S) 42 (W) | 3386 (S) 2276 (W) | 32 (S) 21 (W) |
| DE LEO Diego | Griffith University | Australia | 9 | 355 (S) 520 (W) | 78,765 (S) 88,643 (W) | 81 (S) 77 (W) |
| GUNN Kate | University of South Australia | Australia | 9 | 51 (S) 62 (W) | 706 (S) 677 (W) | 15 (S) 15 (W) |
| BAUMEISTER Harald | ULM University | Germany | 8 | 285 (S) 267 (W) | 7767 (S) 6442 (W) | 46 (S) 42 (W) |

In addition, Figure 6 shows the top 10 authors' productions over time. Figure 6 represents the number of articles by the size of the circles. A larger size means more publications. Moreover, citations per year are represented by the color of the circles. A darker color indicates more citations.

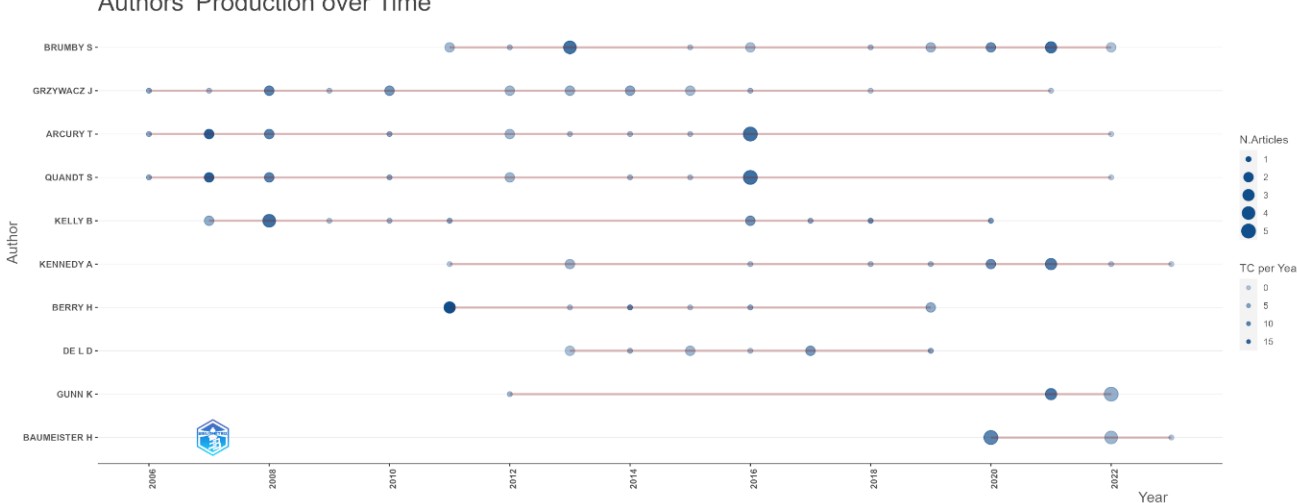

**Figure 6.** Authors' Production over Time.

### 3.2. Keyword Analysis

Figure 7 shows a word cloud created considering the Keywords Plus and their frequency. Most frequent keywords are represented in a larger size. In the setting area, a maximum of 50 words were defined. It is essential to highlight terms like "depression", "agriculture", "stress", and "suicide". It should be noted that relatively recently used terms such as "behavioural health" do not appear in the word cloud of the 50 most frequent terms, as they are not as frequent as the rest.

An overview of the thematic evolution of the topic's scientific publications is shown in Figure 8, which is divided into four periods: 2000–2012, 2013–2018, 2019–2021, and 2022–2023. The focus in the first period is on "farmers", "mental health", with two related concepts, and "farmworker". These concepts evolve into the second period, increasing the importance of "mental health". In this second period, concepts such as "quality of life" appear, directly derived from "farmworker" or "care farming". This is influenced by "mental health problems" and to a lesser extent by "depression" from the previous period.

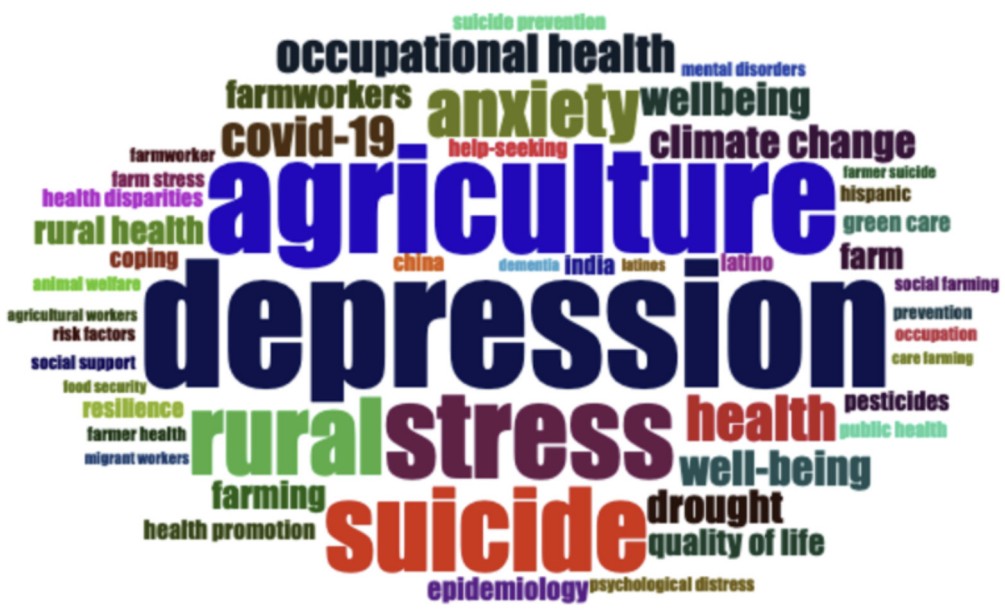

**Figure 7.** Word cloud of the Keywords Plus extracted from both databases analyzed in 2000–2022 using R-studio software and bibliometrix libraries.

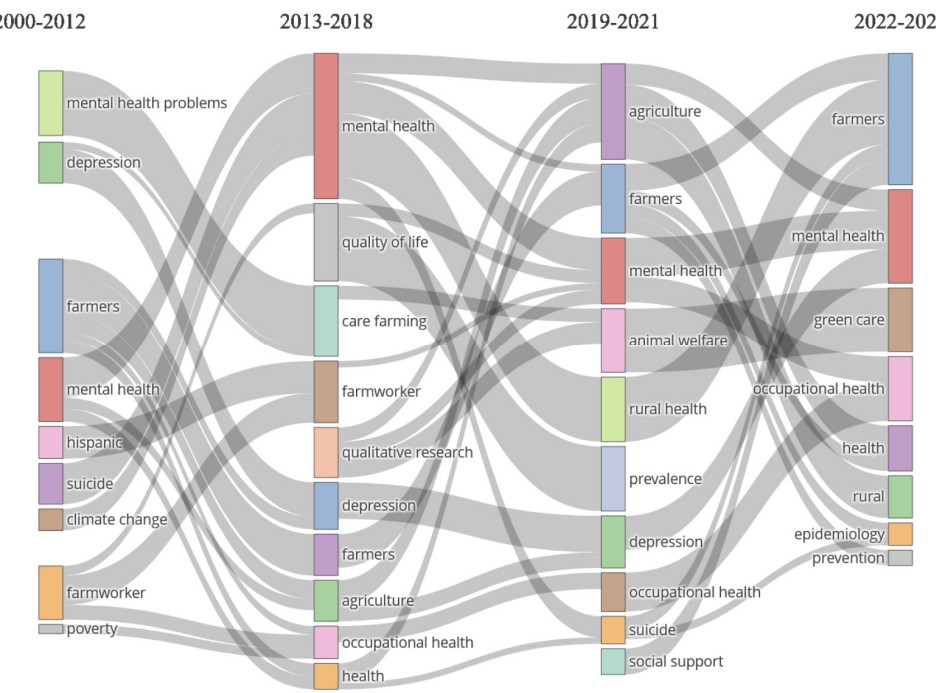

**Figure 8.** Thematic evolution of both databases analyzed using R-studio software and bibliometrix libraries.

In the third period, the concept of "agriculture", which already appeared in the second period, acquires greater significance. Meanwhile, farmers and mental health remain essential concepts. Likewise, "rural health" and "occupational health" appear in this period, illustrating the importance of health.

Finally, in the last period, health is represented by "mental health", "health", and "occupational health". The concept of "farmers" is the most relevant one, showing the importance of research on farmers' physical and mental health.

Figure 9 shows the density of research in each topic related to our study and the external relationship of each topic with the rest of our research topics. These two parameters

define the four areas in the graph. The top left area, called "niche themes", covers the topics with a high density (very much studied internally) but little relationship with the rest of the topics. The upper right area, called "motor themes", is where the density of topics is high (a lot of internal research) and the relationship with the rest of the research topics is powerful. These are the current research topics. The lower left area, called "Emerging or Declining Themes", covers those that have low density and little relation with the rest of the themes, either because their importance in the research is low (declining themes) or because they are very new and have not yet been taken into account for research (emerging themes).

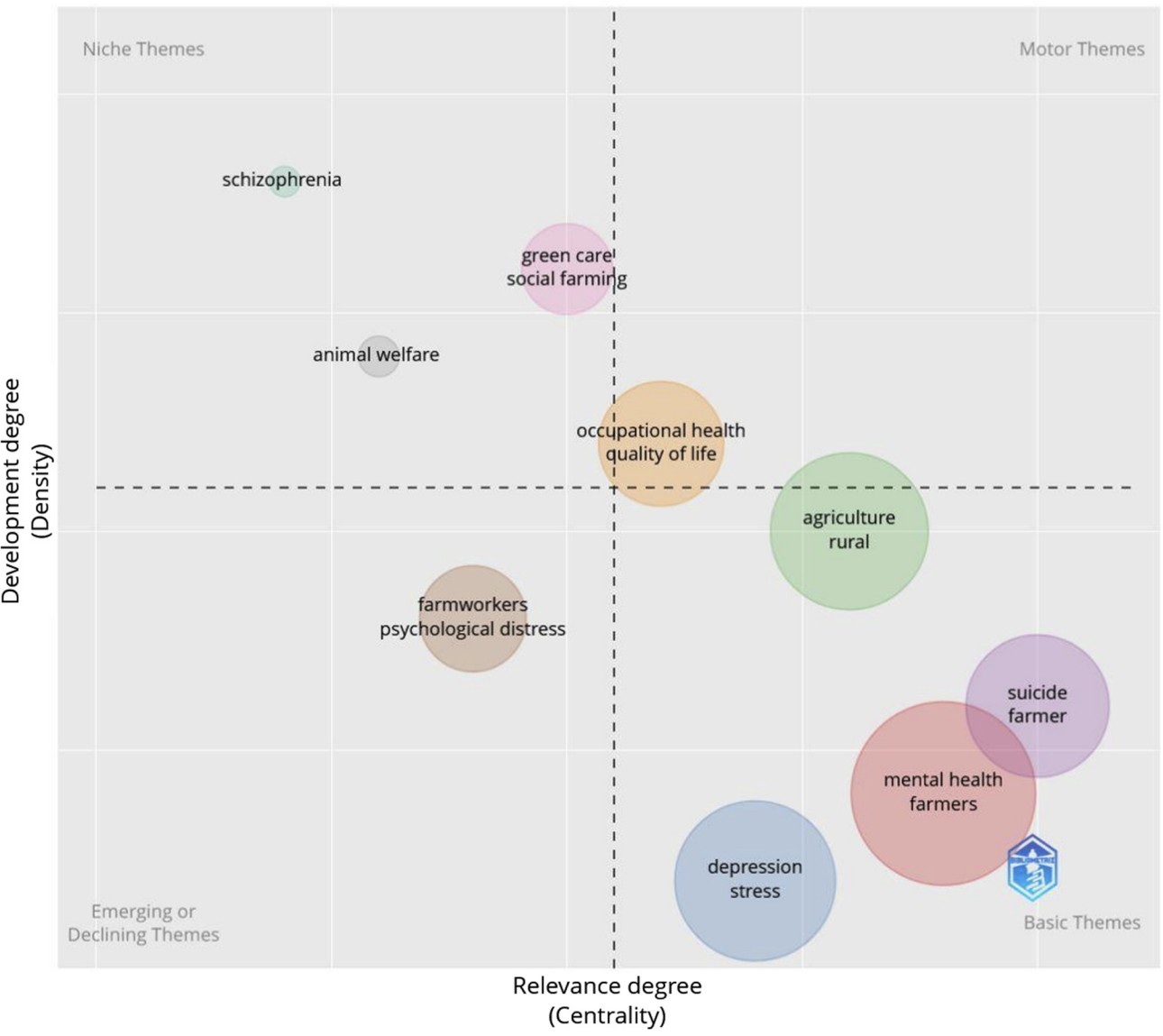

**Figure 9.** Thematic map of both databases analyzed using R-studio and bibliometrix libraries.

Finally, the lower right area ("basic themes") comprises themes with low internal density but which are very much related to the rest of the research themes. These themes must be addressed, as they are relevant to the research and must be well-developed.

Figure 10 shows details of the co-occurrence map with the documents extracted from the Scopus database using VOSViewer. The map shows 9 clusters: "social farming", "risk factors", "quality of life", "mental disorders", "suicide", "prevention", "farmworkers", "agriculture", and "depression". Figure 11 shows the details of the co-occurrence of keywords regarding the "farmers" keyword. This graph illustrates the keywords that

authors affirm to be connected. It highlights words such as: "suicide", "stress", "anxiety", "well-being", "quality of life" and "social support".

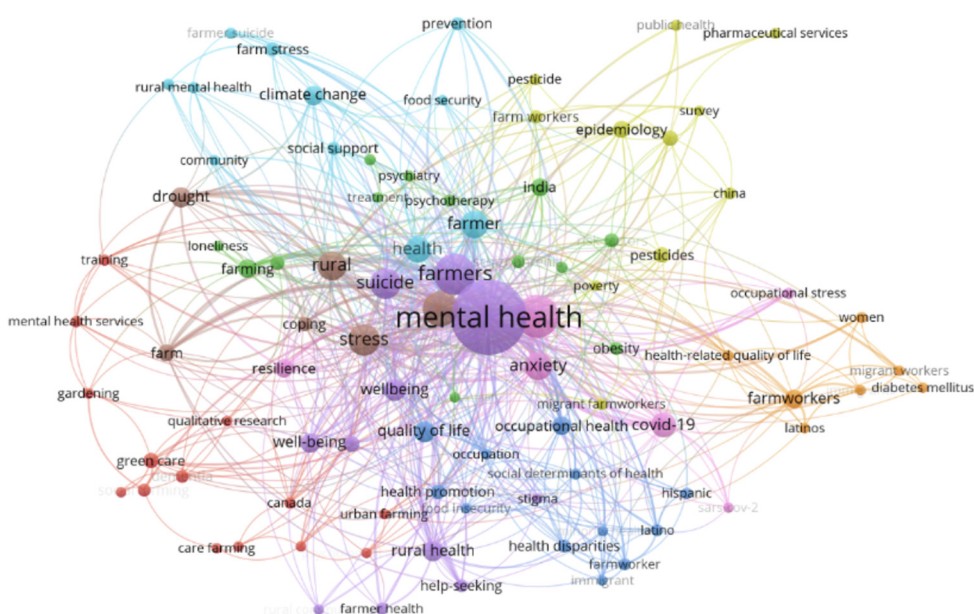

**Figure 10.** Detail of the co-occurrence keywords with both databases during the period 2000–2023 using VOSViewer.

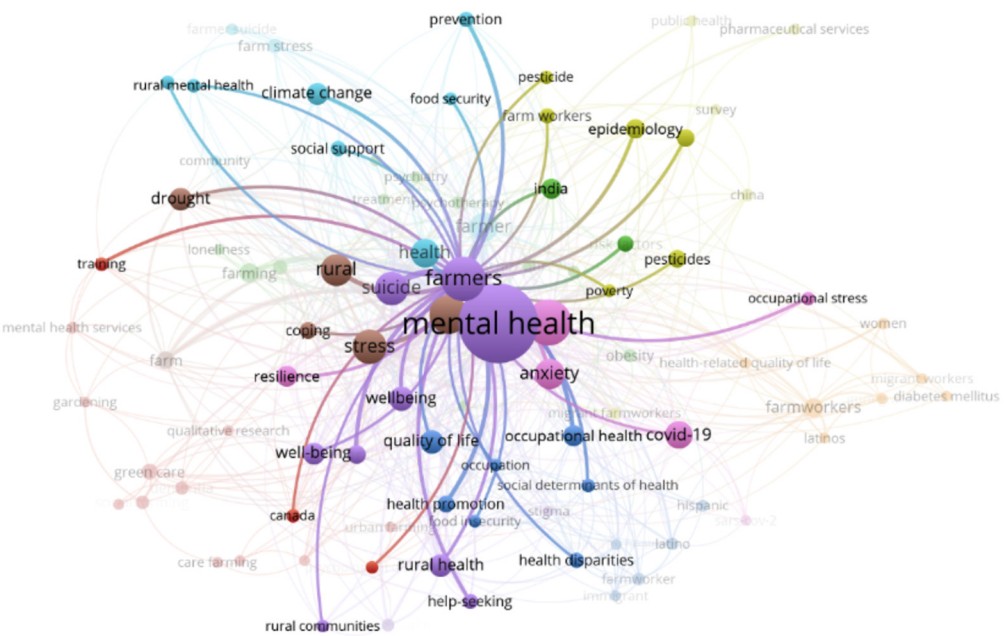

**Figure 11.** Detail of keyword co-occurrence with respect to the keyword "farmers" using VOSViewer.

## 4. Discussion

One of the key aspects related to researchers in all fields of scholarship is keeping updated with relevant literature and findings. The capacity to synthesize research knowledge to boost and contribute to an appropriate line of research [32,33] is a key attribute for researchers in all fields of science. Bibliometric forms are known as a means to provide objective literature, enabling researchers to set studies within the scholarly structure in the field. Moreover, it is accepted that bibliometric methods offer compiled literature

assessments that permit researchers to locate studies in different thematic and intersecting areas of study [33].

The scope of this bibliometric analysis was to review the state of scientific literature about mental health in farming in Scopus and WoS databases. According to the 411 sources, there were 833 documents, 2725 authors, 4014 keywords plus, and 1989 authors' keywords. In addition, the analysis was further enhanced using both Scopus and Web of Science databases.

The review identified five knowledge areas: social science, psychology, agriculture, medicine, and biological sciences. This allowed us to respond to research questions about scientific research on mental health and farmers.

RQ1: What is the quantitative and qualitative level of the scientific research conducted so far on mental health and farmers?

According to the results obtained from the evolution of scientific research on mental health and farming (2000–2022 period), an increase in scientific production from 2017 to 2022 can be stated. Therefore, it is remarkable that a large stream of research on this topic has been boosted during the 21st century. These results are in line with studies in farming about significant changes in terms of technology [15], sharing risk at work [34], low expected profitability [35], remaining competitive in farming [36], constant reallocation of resources in farming [37], or emotional factors related to continuity in family farms [6,8].

RQ2: Which researchers and in which geographical areas have been most investigated on mental health in agriculture?

Our results also confirm that most scientific production on this topic is mainly in the USA and Australia, followed by the UK, Canada, China, and Norway. These former two countries concentrate most of the publications; several authors produced most of the literature about mental health intervention in farmers in these two countries. The social isolation caused by the large geographic extension of countries such as those mentioned, along with Canada or China, may explain the greater awareness and concern of governments and researchers in these countries for the mental health of their farmers. [17]. Moreover, the three affiliations that published the most significant number of documents were the University of South Australia and Deakin University (Australia), the University of Toronto (Canada), and the Medical College of Wisconsin (USA). Similarly, the main authors that published about the topic belong to Australia and the USA, and there are two clusters of working groups, one in each country (Grzywack and Quandt from the USA; Brumby, Gunn and Kennedy from Australia). Remarkably, only one author (Harald from Germany) in the top ten authors in the field of study (2000–2022) does not belong to an Australian or North American university.

RQ3: What are the facets of mental health in agriculture studied in the research carried out so far?

The most used keywords found in the final articles are meaningful ("depression", "agriculture", "stress" and "suicide"), and they show how the main concerns about mental health and farming are identified. Other important keywords are "health", "wellbeing", "rural", "anxiety", "climate change", or "COVID-19". If we delve into the analysis of keyword evolution through the years, as shown in Figure 8, it is interesting to take notice of how "agriculture" evolves into "health promotion" and "agriculture". On the other hand, "mental health" evolves into "agriculture", "social farming", "mental health", "health promotion" and "farmer suicide". Perhaps there is a larger awareness of the connection between "mental health", "health promotion", and even "green care", another word that appears in the last period. This global problem must consider global solutions considering all stakeholders and policymakers. Likewise, in Figure 8, we find a future solution and research line: the evolution to the keywords "social farming" and "health promotion" is an interesting proposal for the farming population and researchers to consider.

If we focus on keywords related to the kind of design or methodology used in the articles, 123 articles used the keyword "survey", 398 used the keyword "questionnaire", and 64 articles used "qualitative research". This is meaningful because when discussing

mental health, the classical quantitative methodology may not be sufficient to provide a deeper understanding of the problems. Along these lines, qualitative articles can show a better and more complete picture, as shown by some authors [9].

Related to keyword analysis, there is evidence about word cloud frequency, highlighting terms like "depression", "agriculture", "stress", and "suicide". Hence, the characteristics that arise from the nature of the occupation [13,38,39], financial standing [40], and other stressors that ensue from the nature of the occupation are key factors in researching mental health in the farming environment. The predominance of these five words results from the fact that life for farmers is more difficult due to globalization's increased economic, environmental, and social demands [41].

RQ4: Which are the emergent research topics related to mental health in farming?

Figure 9 shows that motor and basic themes contribute to the development and consolidation of the research field due to their density and/or centrality. In this case, the figure shows a clear absence of motor themes. Most of the scientific studies on depression, stress, suicide, and mental health use motor theme keywords, which indicates the interest of researchers in these topics. Occupational health and quality of life appear as motor themes, but in a lower position. However, their internal research density is higher than that in the case of the previous themes. Psychological distress is considered an emerging theme as, although it has a low density and centrality, it is close to the central point of the graph. Finally, studies on schizophrenia and animal welfare belong to a group with high density and low centrality and are isolated from the rest of the research on this topic. Ecological care and social farming are in the "niche themes" quadrant but are close to the middle line of centrality. Therefore, although they are in the upper left quadrant, they can be considered relevant to the research.

The findings of this review offer insights for researchers and policymakers about basic themes that represent the backbone of publications (agricultural rural, mental health farmers, depression stress and suicide farmer) and emerging topics such as farmers and psychological distress. Relatedly, topics such as "social farming", "green care", and "schizophrenia" are included as niche themes. Finally, it should be noted that the most central topics were "quality of life" and "occupational health". The findings highlight the need to address research topics and their relevance and emerging topics related to mental health in the farming environment.

In this sense, with the insights from this study, researchers and policymakers can assess these topics and their evolution in farming mental health studies as a guide for showing and identifying knowledge and future research avenues.

*Practical Implications and Future Research*

Recent figures released by international organizations (FAO, EC, OMS) reveal the depth of mental health problems in farm workers in recent years. In this vein, a report entitled "Review on the Future of Agriculture and Occupational Safety and Health (OSH)", commissioned by the European Agency for Safety and Health at Work (EU-OSHA), ranks psychological well-being as a top risk for farmers. For instance, CEJA (European Council of Young Farmers) leads an Erasmus+ project to raise farmers' capacity to cope with mental health issues. Following this line, authors such as [24,42,43] state that the prevalence of mental health disorders among farmers is a growing concern for public health and agricultural authorities.

Based on this study, further research should be conducted on farming populations globally. Nonetheless, while some regions have a vast body of research (especially the USA and Australia), others do not, indicating areas of research gaps that could be attended to. However, the absence of standard parameters that measure mental health problems and their dimensions does not allow for a systematic comparison of the status of this subject among the countries.

Along these lines, the lack of research conducted in Latin America does not necessarily mean the problem of mental health in rural areas does not exist. Rather, it is important

to delve deeper into the subject to identify the state of the problem in Latin America and the Global South. Moreover, national, regional, local, or individual income levels could represent another compelling research inquiry on farmers' mental health. Along with this, the likelihood of accessing financial support or difficulty in obtaining it could mean another research avenue to consider as a stressor in farming.

Other interesting future research lines would be to compare the differences between agricultural policies of the United States, Australia, and even Europe to determine how these differences can affect the mental health of farming workers and help policymakers address this issue. Therefore, it would be an important input for Global South countries where the subject has not yet been addressed profoundly.

The bibliometric review and possible research avenues may prompt future research towards more nuanced insights into the distinctive aspects of mental health in the farming environment and foster the investigation of novel and transversal means in mainstream mental health in farming literature.

## 5. Conclusions

This paper presents with a bibliometric review of articles indexed in Scopus and WoS, allowing us to identify scientific publications on mental health in farming published in the past two decades. In the last ten years, the number of publications has increased exponentially. Most publications in this area come from the USA and Australia, two countries with significant research groups. Brumby at Deakin University in Australia, Grzywacz, and Arcury at Wake Forest University in the USA are the most prolific authors. The keywords most commonly found in the literature are "mental health", "suicide", "stress", and "depression". During the past five years, several topics have evolved. It should be highlighted that the "mental health" keyword evolved into "agriculture", "social farming", "mental health", "health promotion", and "farmer suicide". The emerging topics were "farm workers" and "psychological distress", and the niche themes included "animal welfare", "social farming", and "schizophrenia". It is necessary to emphasize the absence of motor themes in the research on mental health in farmers. As indicated, only the topics "occupational health" and "quality of life" are part of these topics and are very close to the central axes. This indicates the moderate level of scientific production in this field.

On the other hand, there is an essential number of topics closely related to the rest, the basic themes ("suicide", "mental health", "depression", and "stress"), which indicates the scientific interest in these topics within research on mental health in farmers. This review offers an overview of the most relevant research published over the last 20 years. It emphasizes the importance of preventing mental disorders related to isolation and economic losses among farm workers. Notwithstanding the consideration of farmers as an essential part of society (feeding people worldwide) [13], farmers' health studies, especially farmers' mental health, were not performed until the 21st century. However, significant research has been devoted since 2000, revealing an exponential growth of publications related to this topic, especially since 2017. It can be stated that mental health in the farming environment is a theme that is gaining attention, as seen in this review.

In conclusion, the state of mental health in the farming environment can be described as a peak of scientific publications in the last five years. Nowadays, this topic, according to the results obtained, is gaining momentum. This research will permit researchers to take stock of the current direction and ensure that future efforts are undertaken desirably because of the importance of this endeavor and its practical implications for the future of farming.

**Author Contributions:** Conceptualization: M.P.-F. Data curation and Formal analysis: M.D.L. and M.T. Investigation: All the authors. Methodology: M.D.L., M.P.-F. and M.T. Project Administration: M.P.-F. Validation: All the authors. Visualization: M.D.L. and M.T. Writing—original draft: All the authors. Writing—review & editing: All the authors. All authors have read and agreed to the published version of the manuscript.

**Funding:** This research received support from the Family Business Chair of the University de Lleida.

**Data Availability Statement:** All data sources are mentioned in the article.

**Conflicts of Interest:** The authors declare that they have no known competing financial interests or personal relationships that could have appeared to influence the work reported in this paper.

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
