# Peer review of "Mapping the Research into Mental Health in the Farming Environment: A Bibliometric Review from Scopus and WoS Databases"

_agriculture, doi:10.3390/agriculture14010088_

Round 1

Reviewer 1 Report

Comments and Suggestions for Authors

This paper explores a growing field that examines farmers’ mental health and wellbeing. It uses a novel method to do so – bibliometric analysis – and provides important insights to the state of research in this area. The paper needs some work prior to publication. I have provided suggestions below.

Introduction needs attention. Opens with a direct quote that is generic enough in nature to be paraphrased without removing the meaning. There is repetition and redundancy throughout. Structure also needs to be reviewed. Many of the citations are quite old – particularly around mental health problems and suicide rates in farmers – there is a lot of recent literature examining these aspects.

Suggest authors carefully review introduction with a view to remove repetition and redundancy. Consider opening introduction with paragraphs describing the sector and changes in the sector. Consider following this with paragraphs on occupational hazards and risks to mental and emotional wellbeing.

Consider how farming identity might contribute to farmers’ mental and emotional wellbeing and difficulties adapting to changes in the sector.

What is the benefit of a bibliometric review over a systematic literature review or scoping review? What does this add to the literature, or what does this allow you to conclude about the existing literature? Can see that you have done this in the discussion – perhaps consider moving to introduction

In the abstract, authors describe what the paper will do rather than the findings themselves – this can be okay, but consider whether it would be better to describe some of your findings instead.

Section 2.1. uses a variety of past/present/future tense (the bibliometric analysis was/the query will/selection is) – please amend.

Did you screen studies based on whether they were relevant to your research questions? Can see that you excluded duplicates, but after this step there doesn’t appear to be any further screening based on titles/abstracts/full texts. How do you know that all of the included studies were relevant? This might not be a necessary step for bibliometric reviews. 

Page 5, line 166: the University of Toronto is in Canada.

Author Response

Dear Reviewer,

Thank you for recognizing the potential in our work and for giving us positive feedback on the manuscript! In the revision process, closely following the suggestions in the feedback provided by you. We paid special attention to clarifying and elaborating our reasoning in the different aspects you have pointed out. We believe and hope you agree with us that the manuscript has benefitted greatly from this revision. 

Moreover, we have carried out a careful revision and editing of the English language.

Thank you for your precious time.

Reviewer 2 Report

Comments and Suggestions for Authors

Bibliometric analyses are not commonly used in the health sciences so this seems like a novel approach to analyzing the body of work on agricultural stress and mental health. The methods seem appropriate for this type of analysis.

Editing for English grammar and usage is needed in places.

Why was PubMed not included in the databases utilized for this analysis? Why were Web of Science and SCOPUS chosen? Is it because these databases make such an analysis possible by making the data downloadable to software programs to analysis such a large amount of data?

A bibliometric analysis is generally related to citation and impact of journal articles. Could you define how you are using these terms in relation to your research questions?  Isn’t this more about a gap analysis in the literature or an emerging issues assessment?

The terms “behavioral health” which have been commonly employed in recent years do not show up in the word cloud or in the text. Were these terms used in more recent time periods in your results? This term has generally replaced “mental health” when describing rural communities.

Do you think the fact that most studies came out of Australia and the US indicates that there are more concerns about farmer mental health in these countries or is there simply more awareness of these issues in these two countries?

Can you comment on the types of papers that might have been missed with the search terms that were used? For example, there is nothing related to mental health and farm injuries, or substance use. Maybe these other topics were outside the scope of the analysis.

Animal welfare may have become a larger issue due to the avian flu epidemic we have seen in the past few years and the suffering that comes with having to cull a herd.

Comments on the Quality of English Language

English editing is needed.

Author Response

(The authors gave the same response as above.)

Reviewer 3 Report

Comments and Suggestions for Authors

This study presents the intellectual landscape of the mental health in the areas of agriculture. I found the procedure of the analyses is valid and the results make sense to me. However, I have a few concerns about the study. Let me discuss them.

1. Research goals

The research goals are not clear enough in terms of their empirical results. I acknowledge that we need to examine how the studies on mental health of farmers have been done, because we can further understand what topics in agriculture areas should be further investigated. However, the empirical results show which universities actually engaged in the studies or general themes apprearing in publications rather than mental health in details. I expected what kind of mental health has been discussed and what geographical areas have entailed particular mental illnesses. Perhaps, the authors may want to revisit the research questions to match them to the empirical results.

2. Analytic approches vs. visualization 

This paper shows very interesting visualized summaries of the prior research. This is one of the strengths, but if the authors could provide analytic results in addtion to the visualized outcomes, that would be more helpful for readers to understand this study. For example, what frequency/proportional level of each keyword could be presented. Also the correlations between the co-occurrence keywords also help readers understand the whole landscapes. 

That's it. Hope these comments help further develop such a potential work.

Author Response

(The authors gave the same response as above.)
